# ROYAL SOCIETY
# OPEN SCIENCE

Cellular heterogeneity results from indirect effects
under metabolic tradeoffs. *R. Soc. open sci.* **6**:
190281.

**Subject Category:**
Biology (whole organism)

ecology/evolution/synthetic biology

synthetic biology, multicellularity, biofilms,
heterogeneity, mutualism

**Author for correspondence:**
Ricard Solé
e-mail: ricard.sole@upf.edu

Electronic supplementary material is available
online at https://doi.org/10.6084/m9.figshare.c.
4662128.

# Cellular heterogeneity results from indirect effects under metabolic tradeoffs

Aina Ollé-Vila[1,2] and Ricard Solé[1,2,3]

[1]ICREA-Complex Systems Lab, Universitat Pompeu Fabra, 08003 Barcelona, Spain
[2]Institut de Biologia Evolutiva (CSIC-UPF), Psg Maritim Barceloneta, 37,
08003 Barcelona, Spain
[3]Santa Fe Institute, 399 Hyde Park Road, Santa Fe, NM 87501, USA

AO-V, 0000-0002-1411-248X; RS, 0000-0001-6974-1008

The emergence and maintenance of multicellularity requires the coexistence of diverse cellular populations displaying cooperative relationships. This enables long-term persistence of cellular consortia, particularly under environmental constraints that challenge cell survival. Toxic environments are known to trigger the formation of multicellular consortia capable of dealing with waste while promoting cell diversity as a way to overcome selection barriers. In this context, recent theoretical studies suggest that an environment containing both resources and toxic waste can promote the emergence of complex, spatially distributed *proto-organisms* exhibiting division of labour and higher-scale features beyond the cell–cell pairwise interactions. Some previous non-spatial models suggest that the presence of a growth inhibitor can trigger the coexistence of competitive species in an antibiotic-resistance context. In this paper, we further explore this idea using both mathematical and computational models taking the most fundamental features of the *proto-organisms* model interactions. It is shown that this resource-waste environmental context, in which both species are lethally affected by the toxic waste and metabolic tradeoffs are present, favours the maintenance of diverse populations. A spatial, stochastic extension confirms our basic results. The evolutionary and ecological implications of these results are outlined.

## 1. Introduction

Multicellularity has evolved independently at least 25 times in our planet [1–3]. The rise of multicellular systems provided the fabric for further evolutionary events and additional layers of complexity. It became a crucial step towards the rise of complex ecosystems and provided novel opportunities for new niches. It was also a first step towards complex organisms with diverse tissues and organs as well as sensors and actuators. In a different (but deeply related)

**Figure 1.** Multicellular structures in a resource-waste environment. In many different scenarios, heterogeneous microbial populations (*a*) emerge under the presence of external sources of stress limiting their replication potential. Here we show a synthetic, growing population of engineered strains that cooperate under the presence of antibiotics (adapted from Amor *et al.* [10]). In these colonies, the addition of a third 'parasitic' strain that exploits one of the cooperators while degrading the antibiotic can also promote diversity. In a different, but related context, complex *proto-organismal* structures have been obtained (*b*) in an evolutionary spatial model of cell populations with two basic types (yellow and black squares) in a medium (brown sites) including both diffusing nutrients and toxins. The cells evolve specific adhesion patterns that sustain complex, repeatable structures (see Duran-Nebreda *et al.* [13] for details). A mean field model (*c*) aims at capturing the most fundamental features of the PRO model in order to understand the origins of such complex structures. Here $R$ and $T$ stand for resource and toxic components, respectively, while $S_1$ and $S_2$ are the two cellular strains. The toxic $T$ affects both, inhibiting their growth through direct cell mortality with characteristic rates $\gamma_1$ and $\gamma_2$. The non-direct interaction (grey bottom arrow) builds up a positive link which has a critical role in maintaining cellular heterogeneity.

context, most biotransformations that occur in nature are carried out by microbial consortia with different species playing different tasks and often associated in some kind of spatial arrangement (figure 1*a*). In these cases, multicellular assemblies arise as a novel way to deal with environmental sources of stress.

One particular instance that has been widespread is the need to deal with toxic waste, as it can deeply affect the growth potential of cell populations and even determine their decline and extinction. As a result, specialized mechanisms of degradation or neutralization have evolved multiple times. Not surprisingly, important evolutionary transitions in our past biosphere involved dealing with threshold concentrations of toxic or damaging chemicals or radiation levels [4]. Thriving in such selective environments, whether associated physical factors, substrate heterogeneity, shortening of essential nutrients, the presence of toxins or antibiotics, pervades multiple behavioural and self-organizing patterns that require the maintenance of a stable multicellular consortium [5–10]. Likewise, certain trade-offs are also important evolutionary drivers, as has been shown for the trade-off between rate and yield of ATP production [11,12]. In this case, opposite strategies are selected depending on the environmental context, with cooperation and competition having a critical role in the evolutionary outcome.

Models for the origins of MC require a minimal set of assumptions capable of capturing the essential mechanisms required for the maintenance of cell diversity. Such coexistence should be a consequence of cooperative interactions enhancing diversity while providing resilience in the face of environmental fluctuations. In a recent paper [13], it was shown that complex *proto-organismal* (PRO) structures can emerge in a resource-waste environment, as shown in figure 1*b*. In this spatial model, two basic cell types were introduced, linked through a stochastic switching mechanism. One type was able to use resources and replicate, whereas the other was able to process waste at the expense of a reduced replicative potential. Both cell types were affected by the toxic waste through direct cell mortality. The model also incorporated evolutionary dynamics and cell–cell adhesion mediated by an energy minimization rule that plays a significant role in a broad range of morphogenetic phenomena (see [14] and references therein). Despite the lack of a gene regulatory network supporting a true developmental programme, the model was able to generate complex, spatially self-organized patterns far beyond a simple clustered, heterogeneous mixture of cells.

What is the fundamental origin of these *proto-organisms*? What pervades the strong interdependence between cell types? Previous studies on the dynamics of bacterial populations in a chemostat under the presence of antibiotics [15–17] showed that stable coexistence can be achieved under the presence of external inhibitors (mediated by plasmids). Similarly, studies by Eshel Ben-Jacob and co-workers have shown that complex cooperative behavioural patterns (including spatial patterns) emerge out of

cooperative responses to environmental stress [18,19]. These studies reveal the power of self-engineering associated with antibiotic resistance [20]. Here we present a mean field model capturing the most relevant features of the spatial PRO model in order to understand the possible key features predating the observed *proto-organisms*. We find the necessary conditions for the heterogeneous population to exist under the studied scenario, which involve a set of asymmetries and positive correlations allowing us to provide novel insights into the field of toxic-mediated complexity.

## 2. Resource–toxin–consumers model

As described by the diagram in figure 1c, our two cell populations exploit the resource $R$ as consumers while both are damaged by the toxic waste $T$. Since the original inspiration of the model [13] involves closely related cell types, several parameters are taken equal, although as a first approximation we do not link the two cell populations through any developmental nor mutual switching mechanism. The first cell type ($S_1$) replicates at a given rate under the presence of $R$ and is lethally affected by the presence of $T$. On the other hand, the second ($S_2$) can degrade the toxic component $T$ while showing a reduced replication potential. Such trade-off would result from the energy investment associated with $T$ degradation, which necessarily implies that the $S_2$ species will be less efficient at replicating. Importantly, this toxic degrading species also experiences the lethal inhibition from toxic waste $T$.

The full set of equations defining our mean field model are now

$$\frac{dT}{dt} = \mu_T - \delta_T T - \varepsilon S_2 T, \tag{2.1}$$

$$\frac{dR}{dt} = \mu_R - \delta_R R - \eta(S_1 + S_2)R, \tag{2.2}$$

$$\frac{dS_1}{dt} = \rho R S_1 - \gamma_1 T S_1 - \delta_1 S_1 \tag{2.3}$$

and

$$\frac{dS_2}{dt} = \rho(1 - \varepsilon)R S_2 - \gamma_2 T S_2 - \delta_2 S_2, \tag{2.4}$$

where $\mu_{T,R}$ are the influx rates for toxic $T$ and resources $R$, respectively, $\delta_{1,2}$ are the cell mortality rates, $\rho$ is the basal cell replication efficiency, $\varepsilon$ is the efficiency of toxic degradation by $S_2$, which is reflected in a lower efficiency for replication as a trade-off ($\rho(1 - \varepsilon)$), $\eta$ is the rate of resource degradation by both species and $\gamma_{1,2}$ are the mortality rates due to toxic waste for each species.

The previous set of equations has four relevant fixed points ($T^*, R^*, S_1^*, S_2^*$), namely the full extinction point

$$\mathbf{P_1} = (+, +, 0, 0)$$

the coexistence point with both consumers present

$$\mathbf{P_2} = (+, +, +, +)$$

and two single-survivor fixed points, namely:

$$\mathbf{P_3} = (+, +, +, 0) \quad \text{and} \quad \mathbf{P_4} = (+, +, 0, +)$$

(see electronic supplementary material, S-V for the full expressions). Below (and in electronic supplementary material, S-V and S-VI), we study the stability of these points and determine the parametric conditions associated with each.

The model shares some commonalities with the classical one resource-two consumer (1R2C) model [21], where the Competitive Exclusion Principle [22] rules out the possibility of coexisting species. However, the presence of a toxic waste in the medium changes this scenario, creating an indirect positive effect from $S_2$ to $S_1$ (grey line in figure 1c), which rescues the coexistence scenario. We will show how this indirect positive link, the direct cell mortality through toxic waste endured by both cells and the metabolic trade-off of species $S_2$ impose certain constraints in the form of asymmetries and positive correlations that must hold in order to rescue the coexistence scenario.

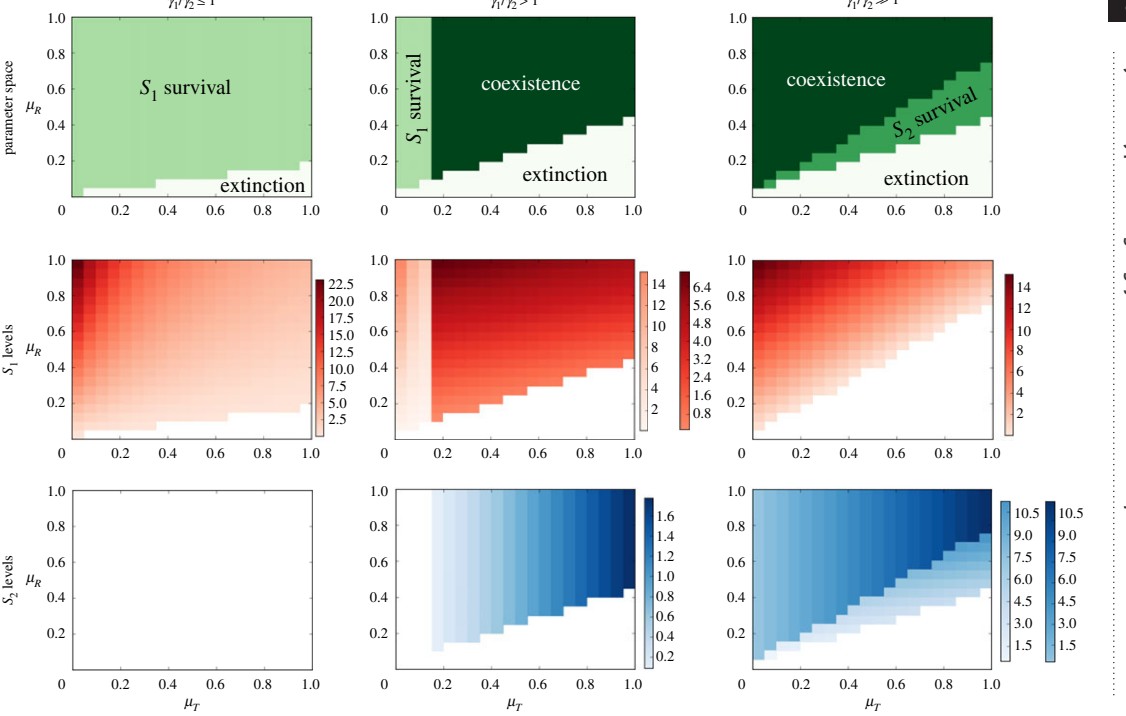

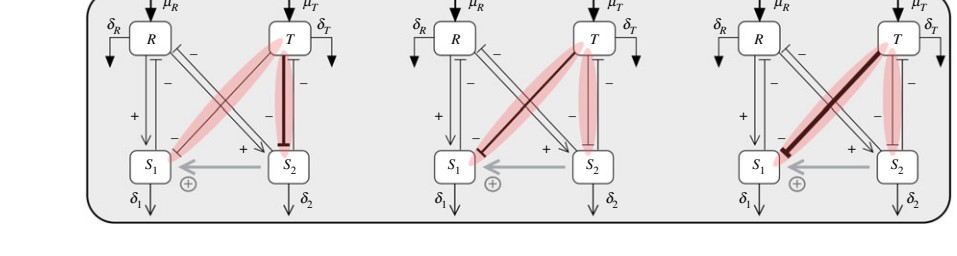

**Figure 2.** Parameter space of the system under different conditions of $\gamma_1/\gamma_2$ (each column) as a function of $\mu_T$ and $\mu_R$. In the first row, we observe how the relative value of these two parameters is determining the absence or the presence of coexistence and regions of $S_1$ and $S_2$ survival. In the second and third rows, we observe the value of the fixed point of $S_1$ and $S_2$, respectively, using different gradients for the different regions of the parameter space shown in the first row. It is observed that an asymmetry in the toxic-mediated cell mortality is needed so that coexistence of $S_1$ and $S_2$ is possible. The values for $R$ and $T$ in the fixed points for these regions can be found in the electronic supplementary material. For $\gamma_1/\gamma_2 \leq 1$, $\gamma_1 = \gamma_2 = 0.1$, for $\gamma_1/\gamma_2 > 1$, $\gamma_1 = 0.17$ and $\gamma_2 = 0.1$ and for $\gamma_1/\gamma_2 \gg 1$, $\gamma_1 = 0.3$ and $\gamma_2 = 0.1$. The rest of the parameters used (all shared in the three parameter spaces) are: $\rho = 0.3$, $\varepsilon = 0.3$, $\eta = 0.1$, $\delta_R = \delta_T = 0.1$, $\delta_1 = \delta_2 = 0.1$.

# 3. Results

## 3.1. Mean field model

The stability analysis of the subsystems $RTS_1$ and $RTS_2$ is properly presented in electronic supplementary material, S-I, II, III leading to a proper intuition to further study the complete system. As the stability analysis of the mean field model is nearly intractable, in order to assess the stability of the presented system we have made use of qualitative stability analysis tools [23–27] (see electronic supplementary material, S-VI). These allow us to show that there are some conditions which do not guarantee the stability of our particular system for any parameter values (see also electronic supplementary material, S-VI for more details). This fact requires a deeper analysis of the system and below we present the conditions that need to hold for our system to be stable. The first is the need for an asymmetry in the toxic sensitivity values $\gamma_{1/2}$ (figure 2). The second demonstrates the critical role of two of the system links which build up an indirect positive relation: these two links need to hold a positive correlation so that coexistence is possible (figure 3a,b). Moreover, our system displays a metabolic trade-off regulated by $\varepsilon$, and this is also constraining the system's behaviour in enabling its full viability (figure 3c). These conditions are fully developed below.

### 3.1.1. Toxic sensitivity asymmetry must hold to observe $S_1$ and $S_2$ coexistence

Although the model involves 11 parameters, just some have been found to be critical in order to understand its behaviour. The parameters determining the influx of $R$ and $T$ ($\mu_R$ and $\mu_T$, respectively) have been found to be significant for identifying regions of the parameter space where coexistence or survival of either of the species is possible (figure 2). Importantly, from an evolutionary perspective, it is quite reasonable to conjecture that the resource and toxic waste levels could be of critical importance for such a system to emerge. Notice that, in general terms, the balance between $\mu_R$ and $\mu_T$ is crucial to determine the coexistence conditions of the species $S_1/S_2$. Similarly, the impact of the toxic waste on each species, weighted by $\gamma_1$ and $\gamma_2$, provides the other major influence.

When $\gamma_1/\gamma_2 = 1$, the impact of the toxic waste is symmetric: it reduces fitness in the same way for both strains. As shown below, the asymmetry between them will be the most relevant trait.

It is important to note that other parameters remain fixed and that a collapse of parameters can be made (see electronic supplementary material, S-VII) without affecting the qualitative behaviour of the system. Resource $R$ and toxic $T$ degradation rates ($\delta_R$ and $\delta_T$) can be collapsed into $\delta$, while cell mortality rates ($\delta_1$ and $\delta_2$) can also be collapsed into a second parameter $\delta_S$. This notation will be used for the next derivations.

In figure 2, we summarize the main results of our analysis. Three particular conditions determine what we observe: (1) $\gamma_1/\gamma_2 \leq 1$, (2) $\gamma_1/\gamma_2 > 1$ and (3) $\gamma_1/\gamma_2 \gg 1$. For the case where we consider $\gamma_1/\gamma_2 > 1$, the actual values of the ratio giving rise to this scenario (second column of figure 2) are approximately 1.5, while for $\gamma_1/\gamma_2 \gg 1$ (third column in figure 2), the actual range of values is approximately [2.5–9]. In the range approximately [1.5–2.5], either coexistence or extinction is observed, but no survival of $S_1$ and $S_2$ on its own.

The asymmetry in these parameters ($\gamma_1 > \gamma_2$) is needed in order to find coexistence in the system. For $\gamma_1 \leq \gamma_2$, coexistence is not possible even if other systems' parameters are changed. The linear boundary separating the presence of cells from its extinction can be easily derived in the latter case. Since $S_2 = 0$, assuming that $R$ and $T$ achieve equilibrium much more rapidly than $S_1$, and thus $\mathrm{d}R/\mathrm{d}t \approx 0$ and $\mathrm{d}T/\mathrm{d}t \approx 0$, we have $T \approx \mu_T/\delta$ and also $R \approx \mu_R/(\delta + \eta S_1) = R(S_1)$. We can write a single-equation model for $S_1$:

$$\frac{\mathrm{d}S_1}{\mathrm{d}t} = \rho R(S_1)S_1 - \Omega S_1 - \delta_S S_1, \tag{3.1}$$

where we define $\Omega = \gamma_1 \mu_T/\delta$ and $R(S_1) = \mu_R/(\delta + \eta S_1)$, respectively. We can see that growth will occur only if $\mathrm{d}S_1/\mathrm{d}t > 0$ or, in other words, if

$$\rho R(S_1) - \Omega - \delta_S > 0, \tag{3.2}$$

which leads, after some algebra, to the threshold condition between $\mu_T$ and $\mu_R$:

$$\mu_R > \phi_1 \mu_T + \phi_2, \tag{3.3}$$

where $\phi_{1,2}$ are combinations of parameters, i.e. $\phi_1 = \gamma_1/\rho$ and $\phi_2 = \delta\delta_S/\rho$ (see electronic supplementary material, S-II for a more detailed analysis). This limit case thus predicts that the input of resources needs to be larger than the one associated with the toxic influx following a simple linear relation (consistently with the phase diagrams at left in figure 2). On the other hand, the larger the negative effect of $T$ (as given by $\gamma_1$) the more difficult for $S_1$ to survive, as expected.

The critical threshold defined by the previous condition is also an estimate of the envelope of the parameter space where populations are expected to be found. The two other columns in the same figure indicate that if $\gamma_1$ is larger than $\gamma_2$ coexistence is achieved.

The original 4-equations model can be reduced, as we did before, to a two-dimensional system (showing the same behaviour as the four-dimensional counterpart, see electronic supplementary material, figures S2 and S3). Once again we assume that the dynamics of both $R$ and $T$ are faster than the ones associated to both $S_1$ and $S_2$ (see electronic supplementary material, S-IV). A system of equations is then obtained, namely

$$\frac{\mathrm{d}S_1}{\mathrm{d}t} = \rho\varphi_1(S_1, S_2)S_1 - \gamma_1\varphi_2(S_2)S_1 - \delta_S S_1 \tag{3.4}$$

and

$$\frac{\mathrm{d}S_2}{\mathrm{d}t} = \rho(1 - \varepsilon)\varphi_1(S_1, S_2)S_2 - \gamma_2\varphi_2(S_2)S_2 - \delta_S S_2. \tag{3.5}$$

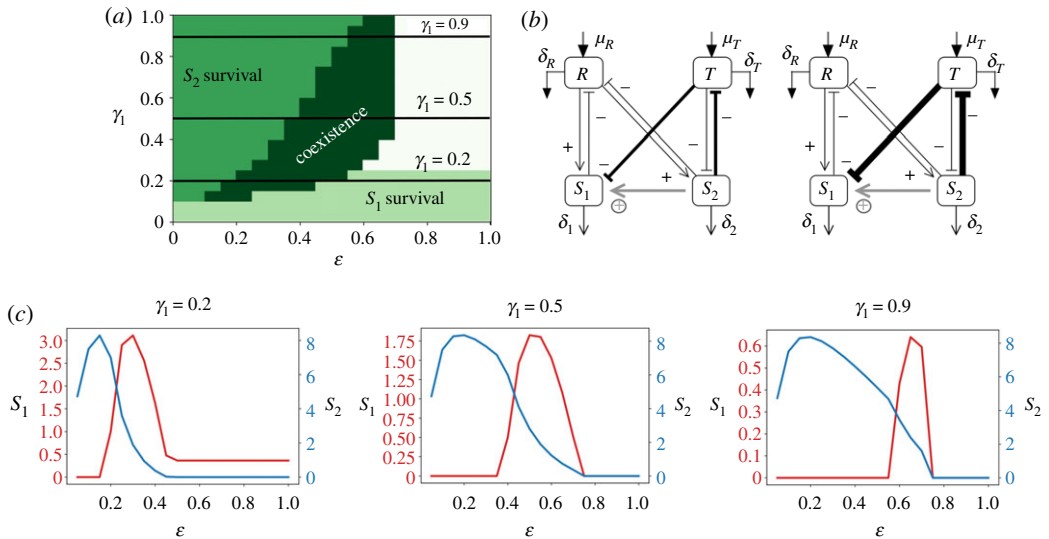

**Figure 3.** Positive correlation between the parameters regulating the efficiency of the indirect positive link from species $S_2$ to $S_1$. (a) Parameter space of the system as a function of $\varepsilon$ and $\gamma_1$ showing the different fixed points the system reaches. (b) System scheme depicting the links of interest that follow a nonlinear positive correlation. (c) Density of species $S_1$ and $S_2$ (red and blue, respectively) for three conditions of $\gamma_1$ (depicted also in a) along the different values of $\varepsilon$ that allows to observe the effect of the metabolic trade-off parameter $\varepsilon$ to the species' fitness. The parameter space shown ($\varepsilon$ versus $\gamma_1$) corresponds to the following constants: $\mu_R = \mu_T = 0.5$, $\gamma_2 = 0.1$, $\delta_1 = \delta_2 = 0.1$, $\rho = 0.3$, $\eta = 0.1$, $\delta_R = \delta_T = 0.1$.

Here the two functions $\varphi_k$ are given by

$$\varphi_1(S_1, S_2) = \frac{\mu_R}{\delta + \eta(S_1 + S_2)} \tag{3.6}$$

and

$$\varphi_2(S_2) = \frac{\mu_T}{\delta + \varepsilon S_2}. \tag{3.7}$$

The two strains will be present provided that the two derivatives are $dS_k/dt > 0$. This leads to two inequalities that need to be simultaneously satisfied, namely:

$$\rho\varphi_1(S_1, S_2) - \gamma_1\varphi_2(S_2) - \delta_S > 0 \tag{3.8}$$

and

$$\rho(1 - \varepsilon)\varphi_1(S_1, S_2) - \gamma_2\varphi_2(S_2) - \delta_S > 0. \tag{3.9}$$

By dividing the two previous equations, it is easy to see that the following inequality is needed, namely we require:

$$\rho\varepsilon\varphi_1 > (\gamma_1 - \gamma_2)\varphi_2 \tag{3.10}$$

and, provided that $\delta_S$ is small enough compared with $S_{1,2}$ the following condition for coexistence condition is found:

$$\mu_R > \left(\frac{\gamma_1 - \gamma_2}{\rho\varepsilon^2}\right)\eta\mu_T. \tag{3.11}$$

Which gives again a linear dependency that will hold only if asymmetry (with $\gamma_1 > \gamma_2$) is present.

Another important feature that can be extracted from figure 2 is that the higher $\gamma_1$ is with respect to $\gamma_2$, the parameter space area with $S_1$ survival vanishes and an area with $S_2$ survival appears. The position of these particular areas is also of importance: $S_1$ can survive on its own as long as $\mu_T$ remains low enough once its mortality rate due to toxic ($\gamma_1$) is already slightly higher than $\gamma_2$. On the other hand, the area with $S_2$ survival appears on the right of the coexistence region: in this region, $\mu_T$ is considerably high and $S_1$ has high sensitivity to $T$, thus only $S_2$ survives.

### 3.1.2. Positive correlation of indirect positive link parameter values

Along with the necessary coexistence condition $\gamma_1 > \gamma_2$, there is an extra condition to be fulfilled. We have found that the indirect positive relationship between $S_1$ and $S_2$ (figure 3) has a major role regulating the viability of the full system. This link has been depicted in figure 1c, formed by the toxic waste degradation by the species $S_2$ ($-\varepsilon T S_2$) and the mortality induced to the species $S_1$ by the toxic waste $T$ ($-\gamma_1 T S_1$).

Taking inequality (3.11), assuming $\gamma_2$ can be neglected for the special case where $\gamma_1 \gg \gamma_2$, we can write

$$\mu_R > \left(\frac{\eta}{\rho}\right)\left(\frac{\gamma_1}{\varepsilon^2}\right)\mu_T \tag{3.12}$$

isolating as a factor in the right-hand side of the equation the term $\gamma_1/\varepsilon^2$ that contains the inhibition $\gamma_1$ of $T$ on $S_1$ as a proportional term along with $\varepsilon$, which acts here as an inverse square. Since $\varepsilon$ is bounded between zero and one, its effects are stronger. This implies that relatively high values of epsilon must be counterbalanced by relatively lower (in comparison) values of $\gamma_1$ in order that the inequality holds, keeping a nonlinear proportionality relationship. Therefore, inequality (3.12) predicts in fact a positive correlation between these two parameters, which thus strongly influences the indirect positive effect.

For the special case $\mu_R = \mu_T$, the critical boundary separating coexistence from other states in the ($\gamma_1$, $\varepsilon$) space will scale as

$$\gamma_1 \sim \left(\frac{\rho}{\eta}\right)\varepsilon^2, \tag{3.13}$$

which of course only provides an (estimated) bound and predicts a nonlinear growth of $\gamma_1$ with $\varepsilon$. However, it is important not to neglect $\gamma_2$ to obtain a complete picture of this relationship. From inequality (3.11), we would obtain

$$\gamma_1 < \varepsilon^2 \left(\frac{\rho}{\eta}\right)\left(\frac{\mu_R}{\mu_T}\right) + \gamma_2, \tag{3.14}$$

with $\gamma_2$ setting the y-intercept of the curve $\gamma_1 = f(\varepsilon)$ as the lower bound below which coexistence is not possible, as the asymmetry $\gamma_1 > \gamma_2$ would be broken. We can observe the good prediction of this equation in electronic supplementary material, figure S19 (section XIII), as well as the effect of $\gamma_2$ constraining the space where coexistence is possible.

In figure 3a, we can observe the actual relationship between $\varepsilon$ and $\gamma_1$ that allows the coexistence of the species. A biological interpretation of this observation could be that, out of the region delimited by $\varepsilon$ and $\gamma_1$, the strength between toxic waste degradation by $S_2$ and the mortality induced by $T$ to $S_1$ becomes too unbalanced, so that the indirect positive effect from $S_2$ to $S_1$ is too weak to actually sustain the survival of both species. We can also see from electronic supplementary material, figure S11 how the positive relationship between $\varepsilon$ and $\gamma_1$ prevails for various combinations of $\mu_R$ and $\mu_T$ values. Notice that if $\varepsilon/\gamma_1 \gg 1$, then $S_2$ is unable to survive anymore due to its poor reproduction capacity (recall that the growth of $S_2$ is regulated by $\rho(1 - \varepsilon)RS_2$), and we find an area of the parameter space of $S_1$ survival. On the other hand, if $\varepsilon/\gamma_1 \ll 1$, then the mortality induced by the toxic waste to $S_1$ is too high for it to survive, and at the same time $S_2$ has a much better reproduction capacity due to lower values of $\varepsilon$, so we find a region of the parameter space of $S_2$ survival.

Figure 3c provides more insights into what is actually happening to the species densities across this parameter space. We can observe that either $S_1$ and $S_2$ have an optimal value of $\varepsilon$ regarding its fitness. The value for $S_2$ is located around 0.2, while the one for $S_1$ is obviously dependent on $\gamma_1$ but also, importantly, on the fitness of $S_2$ associated with $\varepsilon$. Once $\varepsilon$ keeps increasing and $S_2$ degrades toxic with more efficiency and at the same time loses reproduction potential, $S_1$ is able to survive and reach a peak of fitness at a concrete value of $\varepsilon$. In the next section, we extract extra information from this figure regarding the nature of the social relationship between these two species.

With this result, we complete the explanation given above on the conditions for coexistence to be possible: coexistence is possible as long as $\gamma_1 > \gamma_2$ (shown both through the simulations' results in figure 2 and equation 3.11) and, importantly, the efficiencies of the indirect positive link, $\gamma_1$ and $\varepsilon$, follow a nonlinear positive relationship (as observed in the simulation results in figure 3 as well as through equations 3.13 and 3.14). Regarding the effect of other parameter values to the system's behaviour, we have found that an asymmetry in cell mortality rates can lead to a subcritical Hopf bifurcation (see electronic supplementary material, figure S12), but the qualitative behaviour of the system is maintained. Any other change in parameter values does not change the conclusions reached in this paper.

### 3.1.3. Nature of the social interaction between $S_1$ and $S_2$

Taking the parameter spaces studied in the two previous sections ($\mu_T - \mu_R$ and $\varepsilon - \gamma_1$) and performing some extra experiments letting $S_1$ and $S_2$ grow under the same conditions but on their own, we can provide clear insights on the nature of their social interaction (see electronic supplementary material, section XII). We show that species $S_2$ displays a cooperative trait, as the detoxification of a toxic waste can raise the fitness value of its competitor $S_1$. We show how this cooperative trait can be either mutually beneficial or altruistic [28], depending on the considered region of the parameter space.

In electronic supplementary material, figures S17 and S19, we show how the cooperative trait outweighs its costs (balanced by the same parameter $\varepsilon$) for species $S_2$ when grown alone, as its fitness is higher with this trait than in its absence (except above a particular high value of $\varepsilon$, see electronic supplementary material, figure S19). In the case of $S_1$, we can clearly observe how it exploits the cooperative trait displayed by $S_2$, increasing its fitness in the presence of this species compared to when it grows on its own. Moreover, $S_1$ can grow in regions of the parameter space $\mu_R - \mu_T$ that were precluded for it before exploiting $S_2$. Interestingly, as shown in figure 3c, $S_1$ is able to exploit $S_2$ once $\varepsilon$ reaches a certain threshold, but not before ($\gamma_1$ also influences this outcome).

The result of this exploitation is the lower fitness of $S_2$ when grown together with $S_1$ (when they coexist), compared to when it is grown on its own (see electronic supplementary material, figures S17 and S19). We have particularly assessed whether the cooperative trait displayed by $S_2$ is altruistic or mutually beneficial [28]. For the $\mu_R - \mu_T$ space, it is clearly observed that both behaviours are possible depending on the balance of resources and waste (see electronic supplementary material, figure S18), with a necessary positive correlation of the two parameters to observe a cooperative trait that is mutually beneficial. For the case where $\varepsilon$ is studied (see electronic supplementary material, figure S19), we observe that above certain values of $\varepsilon$ the cooperative trait can turn from mutually beneficial to altruistic, as the fitness value of $S_2$ when coexisting with $S_1$ is lower than its basal fitness (considering $\varepsilon = 0$), meaning that the energetic investment in the cooperative trait is no longer beneficial to itself once $S_1$ exploits it.

## 3.2. Discrete stochastic model

The mean field model is validated through the introduction of an spatial counterpart, where stochasticity is present, confirming the robustness of the species coexistence result under the scenario studied in this paper. In order to introduce noise in our model, we have implemented a discrete stochastic version of the mean field model in a $100 \times 100$ lattice. The rules are directly derived from the system of equations defined above, now considered to be discrete:

$$\dot{N}_T = \mu_T - \delta_T N_T - \varepsilon N_{S_2} N_T, \tag{3.15}$$

$$\dot{N}_R = \mu_R - \delta_R N_R - \eta(N_{S_1} + N_{S_2})N_R, \tag{3.16}$$

$$\dot{N}_{S_1} = \rho N_R N_{S_1} - \gamma_1 N_T N_{S_1} - \delta_1 N_{S_1} \tag{3.17}$$

and

$$\dot{N}_{S_2} = \rho(1 - \varepsilon)N_R N_{S_2} - \gamma_2 N_T N_{S_2} - \delta_2 N_{S_2}. \tag{3.18}$$

Each position $(i, j)$ in the lattice contains discrete values of $N_T$, $N_R$, $N_{S_1}$ and $N_{S_2}$ (*reactants*), and they will react locally with the reactants located in the same position $(i, j)$. In this model, there are up to three different types of reactions, depending on how many elements interact: (1) influx of a certain element (just $N_R$ and $N_T$), (2) *one-species* reaction proportional to a certain parameter value and (3) *two-species* reaction when two elements interact also proportional to a particular rate. For a detailed description of the model rules, see electronic supplementary material, S-X and figure S13a. The model includes random dispersal of both strains $S_1$ and $S_2$ using a Moore neighbourhood, as sketched in electronic supplementary material, figure S13b, but does not consider discrete diffusion of $R$ and $T$ molecules.

Comparing the spatial-dependent spaces with those seen in figure 2, it is possible to appreciate some robust traits along with relevant differences. As shown in figure 4 (see also electronic supplementary material, figure S14), species coexistence is always present under local dispersal, in regions precluded in the mean field model. This situation is well known within the field of spatial ecology [29,30]: species coexistence is enhanced by limited diffusion as well as stochasticity. As a final point, the positive nonlinear correlation between $\varepsilon$ and $\gamma_1$ is also confirmed in this stochastic version (see electronic supplementary material, figure S15). The levels of $R$, $T$, $S_1$ and $S_2$ rapidly reach a stable mean value, around which each value of the four system dimensions in every position of the lattice ($i$,

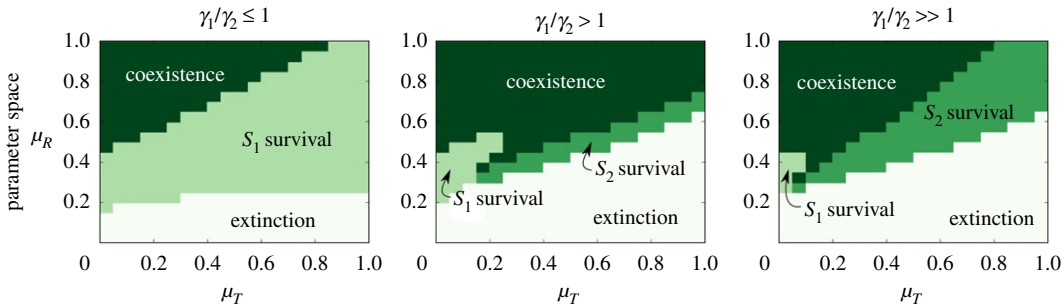

**Figure 4.** Parameter space for the discrete stochastic model under different conditions of $\gamma_1/\gamma_2$ (each column) as a function of $\mu_T$ and $\mu_R$. We observe how the results observed for the numerical simulations in figure 2 are qualitatively recovered, while some changes in the steady state are achieved with this stochastic counterpart in some regions of the parameter space. For $\gamma_1/\gamma_2 \leq 1$, $\gamma_1 = \gamma_2 = 0.1$, for $\gamma_1/\gamma_2 > 1$, $\gamma_1 = 0.17$ and $\gamma_2 = 0.1$ and for $\gamma_1/\gamma_2 \gg 1$, $\gamma_1 = 0.3$ and $\gamma_2 = 0.1$.

$j$) oscillates (see electronic supplementary material, figure S16 for a sample of the temporal evolution of the mean values $\pm$ s.d. of each dimension).

## 4. Discussion

In this paper, we explore the persistence of stable heterogeneous cell assemblies exploiting a common resource while exposed to a toxic waste. This situation is relevant to both the evolution of multicellularity and the maintenance of heterogeneous populations (as it occurs with biofilms [31]). In our system, one strain is capable of removing the toxic waste at the expense of a reduced reproductive potential under a metabolic trade-off. Similar scenarios have been previously studied, from mean field analysis [15–17] to agent-based models [32,33] as well as *in vitro* and *in vivo* experiments [33–37]. All of them have in common that a resistant species detoxifies the medium from a specific antibiotic, while in some of them a payoff in fitness is paid by the species displaying the detoxifying mechanism.

A recent theoretical work on the emergence of '*proto-organisms*' (PRO) exhibiting division of labour and higher-scale features [13] considers a different scenario with respect to the aforementioned models. In that model, the detoxifying species is also sensible to the toxic waste it is degrading, while the metabolic trade-off is characterized in a previously unconsidered way, as the same parameter $\varepsilon$ balances the detoxifying power of species $S_2$ while inhibiting its reproductive capacity with the same strength. These differential features are now taken into account in the mean field model. Beyond the differences with the original PRO model (agent-based model including evolvable differential cell adhesion and stochastic phenotypic switching), we take the most fundamental features of it in order to understand the possible key features predating the observed complex PRO.

In our model, a characterization of the strains is introduced through the sensitivities ratio ($\gamma_1/\gamma_2$). To our knowledge, this is a previously unstudied scenario, in which no assumption is made regarding the sensitivity to a toxic waste either of the toxic-degrading species or its resource-competitor. Within the context of cancer, genetic and phenotypic heterogeneity are an essential part of the somatic evolution of tumours [38,39], while in a study where a microbial oxygen reactive species sink is proved to provide a milder niche to survive to a very common phytoplankton species [37], it is found that certain ecotypes of phytoplankton still carry low efficiency mechanisms of oxidative stress tolerance. These are examples of systems that our model can better represent by explicitly including varying degrees of sensitivity/resistance to toxic waste by the interacting species.

Our analysis provides well defined, robust predictions regarding stability. For our network of species–resource–toxic interactions, well-defined domains of coexistence are found. We show how the asymmetry in toxic sensitivities either breaks or allows the coexistence of the species, while we also generalize the obtained results in the resource-waste inputs parameter space using the sensitivities ratio. In particular, the sensitivity to toxic waste for the toxic-degrading species $S_2$ must be lower than its resource–competitor $S_1$ so that coexistence is possible. In an evolutionary context, according to our mean field model, this would imply that a putative mutation into a new species capable of degrading a certain toxic waste should display a lower sensitivity to it (compared to the *wild type* species) in order to allow coexistence of the two species.

The indirect positive relation from the toxic degrading species (cooperator) to its resource–competitor species has been found to be critical for heterogeneity. Moving beyond previous work, we have found

that a positive nonlinear correlation between the efficiency parameters (regulating the indirect link) needs to hold to allow coexistence. Moreover, we have also observed the effect of the sensitivity to toxic waste of the cooperator species in this interaction, precluding domains of species coexistence. To our knowledge, no previous related work had characterized to this level the cooperative link from $S_2$ to $S_1$, which here is also modified by the fact that both species are sensible to the toxic waste.

The spatial stochastic model confirms the robustness of the previous results. Related works argue that the intermixing between a sensitive and a resistant species is needed so that the detoxifying effect of the latter can have a significant impact on the former to avoid its extinction [32,34]. In this context, Estrela *et al.* [32] argue that this interaction will only be stable if both species display a mutualistic relationship through cross-feeding metabolites, although it has also been shown that in spite of the presence of cross-feeding interactions which are mutually beneficial, competition or exploitation are also possible outcomes—while unlikely in terms of its evolutionary stability due to mechanistic constraints [40]. Sorg *et al.* [34] proposed that the competition exclusion principle [22] can be avoided through strong ecological feedback, niche partitioning or certain spatio-temporal structures, without a need to rely on mutualistic interactions. Our spatial model, enabling more than one cell at each site, enhances (although it does not ensure) species intermixing, and the root of its stability is putative to be in the lines of the aforementioned works.

In any case, the spatial dimension has been proved to have a strong impact on the outcome of these kind of interactions compared with liquid culture assays [33]. Importantly, the spatial model displays a marked difference with respect to the mixed case: the asymmetry of toxic sensibilities is no longer needed to find coexistence. There is a domain in our parameter space where the sensibility to toxic waste of the competitor was lower than the cooperative species and they were still able to coexist. This poses an interesting scenario that would require further work to properly analyse it, by explicitly including spatial dimensions under a reaction–diffusion framework.

Regarding the possibility that the presented system is evolutionarily stable, further work would be needed to assess it. We have shown that the cooperative trait displayed by the detoxifying species can either be mutually beneficial or altruistic [28], depending on the levels of both resource and waste. Therefore, our system would *a priori* have more chances of being evolutionarily stable in the regions where the cooperative trait is beneficial for both resource competitors. Despite the difficulty in explaining altruism [28], some studies have proved that it can be strongly selected in special environments (such as under the presence of heavy metals), in spite of the huge exploitation it entails [35]. Other *in vitro* works have shown that interspecies collective resistance is possible [34], while sensitive species can outcompete resistant ones in determinate conditions [33,34], compromising the stability of the interaction. A particular *in vitro* experiment proved coexistence to be difficult, finding it only for high densities of sensitive species when behaving as persister cells exploiting the resistant detoxifying ones in the long term [36].

Nevertheless, social interactions that can be mapped to the snowdrift game, in which when all your neighbours are defecting it is best to cooperate, might explain why a strategy of this kind can be evolutionarily stable through negative frequency dependence [36]. The goal of the present work was to elucidate the most fundamental features predating the complex structures found in the PRO work [13]. Therefore, we can argue that the evolvable differential adhesion and the stochastic phenotypic switching introduced in the PRO model could be the sufficient ingredients to ensure a formerly unstable social relationship (with public goods and cheaters) could evolve into a stable one. Although a full, explicit evolutionary model following adaptive dynamics and changing rates is out of the scope of this paper, it will be studied in future work in order to provide deeper insights filling the gap between our work and the PRO one, assessing if less conditions would be sufficient to ensure the evolutionary stability of the relationship we have thoroughly described in the present work.

Data accessibility. The results shown have been generated through the widely known Runge–Kutta numerical method (see electronic supplementary material, S-XI) while the Discrete Stochastic model can be easily reproduced with the information given in the paper (section III-B in main text and electronic supplementary material, S-X). The linear stability analyses of the four-dimensional system, of its two-dimensional counterpart and of the three-species subsystems ($RTS_1$ and $RTS_2$) are included in the electronic supplementary material, S-I to V. The qualitative stability analysis of the four-dimensional system is included in electronic supplementary material, S-VI. A collapse of parameters used in the derivation shown in the main text is included in electronic supplementary material, S-VII. A further assessment of the positive correlation of the parameters regulating the efficiency of the indirect positive interaction is shown in electronic supplementary material, S-VIII, where it is shown to be maintained for different values of $\mu_R$ and $\mu_T$. The Hopf bifurcation found for a specific parameter region is shown in electronic supplementary material, S-IX. As mentioned, a detailed description of the Discrete Stochastic model

implementation is found in electronic supplementary material, S-X, through which it can be easily reproduced. In this same section, we also show the parameter space presented in the main text additionally including the $S_1$ and $S_2$ levels, as well as how it is maintained the positive correlation between the parameters regulating the efficiency of the indirect positive interaction and the temporal evolution of the four reactants in this discrete stochastic spatial model. As mentioned, the numerical method used is described in electronic supplementary material, S-XI. A further analysis on the nature of the social interaction of species $S_1$ and $S_2$ is included in electronic supplementary material, S-XII. Finally, the effect of the toxic sensitivity of the cooperative species $S_2$ to the positive nonlinear correlation between the parameters regulating the efficiency of the indirect positive interaction is shown in electronic supplementary material, S-XIII.

Authors' contributions. R.S. conceived the idea and design of this work, derived the analytic results, analysed and interpreted results and wrote the article. A.O.V. helped in the design of the work, developed the code, performed the experiments, analysed and interpreted results and wrote the article.

Competing interests. There are no competing interests in this work.

Funding. A.O.V. received funding from Universities and Research Secretariat of the Ministry of Business and Knowledge of the Generalitat de Catalunyaand the European Social Fund. R.S. received funding from the Botín Foundationby Banco Santander through its Santander Universities Global Division, a MINECO grant no. FIS2015-67616 fellowship co-funded by FEDER/UE and from the Santa Fe Institute.

Acknowledgements. The authors thank the members of the Complex Systems Lab for useful discussions. Special thanks to Rosa Martínez-Corral and Jordi Piñero for our discussions and their valuable suggestions and to G. Taro for her inspiring ideas.

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
