## [Reviewer comments · Royal Society Open Science]

Review History

RSOS-190281.R0 (Original submission)

Review form: Reviewer 1

Is the manuscript scientifically sound in its present form?

No

Are the interpretations and conclusions justified by the results?

Yes

Is the language acceptable?

No

Is it clear how to access all supporting data?

Not Applicable

Do you have any ethical concerns with this paper?

No

Have you any concerns about statistical analyses in this paper?

No

Recommendation?

Major revision is needed (please make suggestions in comments)

Comments to the Author(s)

In "Cellular heterogeneity results from indirect effects under metabolic tradeoffs" the authors consider a toxin-resource model with two species and analyse the parameter regimes that lead to coexistence and exclusion. The key difference between the two species is that one can detoxify the environment at a cost to its own fecundity. The authors identify that an asymmetry in toxin susceptibility between the species is particularly important in maintaining co-existence. They link their work to the origins of multicellularity as well as the maintenance of heterogeneous microbial populations.

The topic of this paper is particularly interesting. The model(s) and the analyses seem to be of acceptable quality for the journal. My chief concern is with the novelty of the work. There have been many previous investigations into the effects of toxins/stress and tradeoffs in maintaining diversity. The authors leave out many relevant works that would better establish what elements of the paper are actually novel. For example, here is a brief list of some relevant papers:

S Estrela ("Community interactions and spatial structure shape selection on antibiotic resistant lineages")

S O'Brien ("Social evolution of toxic metal bioremediation in *Pseudomonas aeruginosa*")

S Estrela ("From metabolism to ecology: cross-feeding interactions shape the balance between polymicrobial conflict and mutualism")

HJ Goldsby ("The Evolutionary Origin of Somatic Cells under the Dirty Work Hypothesis")

JJ Morris ("Dependence of the Cyanobacterium *Prochlorococcus* on Hydrogen Peroxide Scavenging Microbes for Growth at the Ocean's Surface")

HJ Goldsby ("Task-switching costs promote the evolution of division of labor and shifts in individuality")

RA Sorg ("Collective Resistance in Microbial Communities by Intracellular Antibiotic Deactivation")

I Frost ("Cooperation, competition and antibiotic resistance in bacterial colonies")

F Medaney ("Live to cheat another day: bacterial dormancy facilitates the social exploitation of β -lactamases")

My concern with the novelty is not simply a suggestion that the authors cite more papers. I believe that the asymmetry in toxin susceptibility that was identified to be important may recapitulate the results of other models that explore the co-existence between toxin-susceptible and toxin-resistant types. Also, the metabolic tradeoffs that the authors explore are implemented as a difference in the population growth rates between the two species which is also fairly common in models with toxin-susceptible and toxin-resistant types. Thus, it is not clear to me what is new in the paper.

I think the links to evolution also need to be improved if they are going to be used to justify the significance of the work. I do not think the work has any connection to the evolution of multicellularity as there is no mechanism whereby groups of cells can give rise to new groups. I would forego the tenuous multicellularity connection in favour of the connection to microbial consortia which seems more plausible based on the model(s). However, if the maintenance of diversity in a microbial community is the main focus of this work then one aspect that should be considered is what if the functional roles of types can evolve. This is a key aspect of many papers that study microbial social evolution and the maintenance of diversity/cooperation. Under the parameter regimes in the model(s) with co-existence steady states, are the steady states also

evolutionary stable? In other words, if the detoxifying types could mutate to increase/decrease their detoxification rate, what happens? I think that this point is particularly relevant to the maintenance of diversity.

Decision letter (RSOS-190281.R0)

14-May-2019

Dear Miss Ollé-Vila,

The editors assigned to your paper ("Cellular heterogeneity results from indirect effects under metabolic tradeoffs") have now received comments from reviewers. We would like you to revise your paper in accordance with the referee and Associate Editor suggestions which can be found below (not including confidential reports to the Editor). Please note this decision does not guarantee eventual acceptance.

Please submit a copy of your revised paper before 06-Jun-2019. Please note that the revision deadline will expire at 00.00am on this date. If we do not hear from you within this time then it will be assumed that the paper has been withdrawn. In exceptional circumstances, extensions may be possible if agreed with the Editorial Office in advance. We do not allow multiple rounds of revision so we urge you to make every effort to fully address all of the comments at this stage. If deemed necessary by the Editors, your manuscript will be sent back to one or more of the original reviewers for assessment. If the original reviewers are not available, we may invite new reviewers.

- Data accessibility

It is a condition of publication that all supporting data are made available either as supplementary information or preferably in a suitable permanent repository. The data accessibility section should state where the article's supporting data can be accessed. This section

should also include details, where possible of where to access other relevant research materials such as statistical tools, protocols, software etc can be accessed. If the data have been deposited in an external repository this section should list the database, accession number and link to the DOI for all data from the article that have been made publicly available. Data sets that have been deposited in an external repository and have a DOI should also be appropriately cited in the manuscript and included in the reference list.

If you wish to submit your supporting data or code to Dryad (<http://datadryad.org/>), or modify your current submission to dryad, please use the following link:
<http://datadryad.org/submit?journalID=RSOS&manu=RSOS-190281>

- **Competing interests**

- **Authors' contributions**

- **Acknowledgements**

- **Funding statement**

on behalf of Dr Punidan Jeyasingh (Associate Editor) and Kevin Padian (Subject Editor)
openscience@royalsociety.org

Associate Editor's comments (Dr Punidan Jeyasingh):

I apologize for the long delay. I had an unusually difficult time finding suitable and willing reviewers for this manuscript. Of the 18 invited, only 1 accepted to review. As such, I am forced to rely on one review and my own (non specialist) review of the manuscript. I felt the manuscript was well-written, although I had to make an effort to glean the ecological and evolutionary implications of the work. The expert reviewer is uniquely qualified in this area, and returned a thoughtful and constructive review. The reviewer felt that the manuscript does not harness prior work in the area. Furthermore, the authors have to flesh out the evolutionary context of the work better, or focus only on ecological implication, as the reviewer suggests. While I am not too concerned with novelty of the work, prior work in the area needs to be properly acknowledged and inferences from the current work discussed/interpreted with discoveries of such prior work. With much gratitude to the expert reviewer, I invite the authors to submit a revised version addressing these issues.

Associate Editor: 2

Comments to the Author:

(There are no comments.)

Comments to Author:

Reviewers' Comments to Author:

Reviewer: 1

Comments to the Author(s)

In "Cellular heterogeneity results from indirect effects under metabolic tradeoffs" the authors consider a toxin-resource model with two species and analyse the parameter regimes that lead to coexistence and exclusion. The key difference between the two species is that one can detoxify the environment at a cost to its own fecundity. The authors identify that an asymmetry in toxin susceptibility between the species is particularly important in maintaining co-existence. They link their work to the origins of multicellularity as well as the maintenance of heterogeneous microbial populations.

The topic of this paper is particularly interesting. The model(s) and the analyses seem to be of acceptable quality for the journal. My chief concern is with the novelty of the work. There have been many previous investigations into the effects of toxins/stress and tradeoffs in maintaining diversity. The authors leave out many relevant works that would better establish what elements of the paper are actually novel. For example, here is a brief list of some relevant papers:

S Estrela ("Community interactions and spatial structure shape selection on antibiotic resistant lineages")

S O'Brien ("Social evolution of toxic metal bioremediation in *Pseudomonas aeruginosa*")

S Estrela ("From metabolism to ecology: cross-feeding interactions shape the balance between polymicrobial conflict and mutualism")

HJ Goldsby ("The Evolutionary Origin of Somatic Cells under the Dirty Work Hypothesis")

JJ Morris ("Dependence of the Cyanobacterium *Prochlorococcus* on Hydrogen Peroxide Scavenging Microbes for Growth at the Ocean's Surface")

HJ Goldsby ("Task-switching costs promote the evolution of division of labor and shifts in individuality")

RA Sorg ("Collective Resistance in Microbial Communities by Intracellular Antibiotic Deactivation")

I Frost ("Cooperation, competition and antibiotic resistance in bacterial colonies")

F Medaney ("Live to cheat another day: bacterial dormancy facilitates the social exploitation of β -lactamases")

My concern with the novelty is not simply a suggestion that the authors cite more papers. I believe that the asymmetry in toxin susceptibility that was identified to be important may recapitulate the results of other models that explore the co-existence between toxin-susceptible and toxin-resistant types. Also, the metabolic tradeoffs that the authors explore are implemented as a difference in the population growth rates between the two species which is also fairly common in models with toxin-susceptible and toxin-resistant types. Thus, it is not clear to me what is new in the paper.

I think the links to evolution also need to be improved if they are going to be used to justify the significance of the work. I do not think the work has any connection to the evolution of multicellularity as there is no mechanism whereby groups of cells can give rise to new groups. I would forego the tenuous multicellularity connection in favour of the connection to microbial consortia which seems more plausible based on the model(s). However, if the maintenance of diversity in a microbial community is the main focus of this work then one aspect that should be considered is what if the functional roles of types can evolve. This is a key aspect of many papers that study microbial social evolution and the maintenance of diversity/cooperation. Under the parameter regimes in the model(s) with co-existence steady states, are the steady states also evolutionary stable? In other words, if the detoxifying types could mutate to increase/decrease their detoxification rate, what happens? I think that this point is particularly relevant to the maintenance of diversity.

Author's Response to Decision Letter for (RSOS-190281.R0)

See Appendix A.

RSOS-190281.R1 (Revision)

Review form: Reviewer 1

Is the manuscript scientifically sound in its present form?

Yes

Are the interpretations and conclusions justified by the results?

Yes

Is the language acceptable?

Yes

Do you have any ethical concerns with this paper?

No

Have you any concerns about statistical analyses in this paper?

No

Recommendation?

Accept as is

Comments to the Author(s)

Satisfied.

Decision letter (RSOS-190281.R1)

20-Aug-2019

Dear Miss Ollé-Vila,

I am pleased to inform you that your manuscript entitled "Cellular heterogeneity results from indirect effects under metabolic tradeoffs" is now accepted for publication in Royal Society Open Science.

on behalf of Dr Punidan Jeyasingh (Associate Editor) and Kevin Padian (Subject Editor)
openscience@royalsociety.org

Associate Editor Comments to Author (Dr Punidan Jeyasingh):

I thank the authors for their detailed revisions in response to the reviewer comments. This version reads much better and clearly the expert reviewer is also satisfied. I am happy to recommend publication.

Reviewer comments to Author:

Reviewer: 1

Satisfied.

Appendix A

Editor's comments

"Associate Editor's comments (Dr Punidan Jeyasingh):

I apologize for the long delay. I had an unusually difficult time finding suitable and willing reviewers for this manuscript. Of the 18 invited, only 1 accepted to review. As such, I am forced to rely on one review and my own (non specialist) review of the manuscript. I felt the manuscript was well-written, although I had to make an effort to glean the ecological and evolutionary implications of the work."

RESPONSE: We thank the editor for all the efforts to make the referee/editorial work possible and understand the difficulties associated with interdisciplinary papers like ours, specially nowadays with the huge amount of submissions that make the process so tangled. We have indeed addressed your (and referee's) comment on the need of a better discussion targeting evolutionary and ecological issues.

"The expert reviewer is uniquely qualified in this area, and returned a thoughtful and constructive review. The reviewer felt that the manuscript does not harness prior work in the area. Furthermore, the authors have to flesh out the evolutionary context of the work better, or focus only on ecological implication, as the reviewer suggests."

RESPONSE: Indeed we are happy with the referee report, which raises relevant points concerning previous literature and potential shortcomings that we have addressed in the revised version. We hope the evolutionary context is now clearer while we have added some extra analysis that better explain the social interaction between the species, focusing on its ecological implications.

"While I am not too concerned with novelty of the work, prior work in the area needs to be properly acknowledged and inferences from the current work discussed/interpreted with discoveries of such prior work."

RESPONSE:

Some prior work in the area was acknowledged in the first version of our manuscript, particularly in relation with Lenski paper and related ones. However, the additional references brought by the reviewer will certainly improve the quality and breath of those previous works that connect with ours. We have included them in the revised version.

"With much gratitude to the expert reviewer, I invite the authors to submit a revised version addressing these issues."

RESPONSE:

Thanks again; we have done our best (with the constraints imposed by the paper limits and our original goals) to address all the comments and queries. All the changes added in the paper are appropriately highlighted in green, either in the main text and the supplementary material.

Associate Editor: 2

Comments to the Author:

(There are no comments.)"

Reviewer 1 comments

"Reviewers' Comments to Author:

In "Cellular heterogeneity results from indirect effects under metabolic tradeoffs" the authors consider a toxin-resource model with two species and analyse the parameter regimes that lead to coexistence and exclusion. The key difference between the two species is that one can detoxify the environment at a cost to its own fecundity. The authors identify that an asymmetry in toxin susceptibility between the species is particularly important in maintaining co-existence. They link their work to the origins of multicellularity as well as the maintenance of heterogeneous microbial populations."

RESPONSE: We appreciate the referee's critical revision of our paper. As discussed below, we have addressed the comments and queries concerning the need of a better representation of our work and its novelty and particularly the proper approach to the evolutionary implications of the work. We now also provide deeper insights into the nature of the social interaction between the competitor species. All the changes added in the paper are appropriately highlighted in green, either in the main text and the supplementary material.

"The topic of this paper is particularly interesting. The model(s) and the analyses seem to be of acceptable quality for the journal. My chief concern is with the novelty of the work. There have been many previous investigations into the effects of toxins/stress and tradeoffs in maintaining diversity. The authors leave out many relevant works that would better establish what elements of the paper are actually novel. For example, here is a brief list of some relevant papers:

S Estrela ("Community interactions and spatial structure shape selection on antibiotic resistant lineages")

*S O'Brien ("Social evolution of toxic metal bioremediation in *Pseudomonas aeruginosa*")*

S Estrela ("From metabolism to ecology: cross-feeding interactions shape the balance between polymicrobial conflict and mutualism")

HJ Goldsby ("The Evolutionary Origin of Somatic Cells under the Dirty Work Hypothesis")

*JJ Morris ("Dependence of the Cyanobacterium *Prochlorococcus* on Hydrogen Peroxide Scavenging Microbes for Growth at the Ocean's Surface")*

HJ Goldsby ("Task-switching costs promote the evolution of division of labor and shifts in individuality")

RA Sorg ("Collective Resistance in Microbial Communities by Intracellular Antibiotic Deactivation")

I Frost ("Cooperation, competition and antibiotic resistance in bacterial colonies")

F Medaney ("Live to cheat another day: bacterial dormancy facilitates the social exploitation of β -lactamases")

My concern with the novelty is not simply a suggestion that the authors cite more papers. I believe that the asymmetry in toxin susceptibility that was identified to be important may recapitulate the results of other models that explore the co-existence between toxin-susceptible and toxin-resistant types. Also, the metabolic tradeoffs that the authors explore are implemented as a difference in the population growth rates between the two species which is also fairly common in models with toxin-susceptible and toxin-resistant types. Thus, it is not clear to me what is new in the paper."

RESPONSE: As the referee knows, we are aware of the existence of related works and some were already cited in the paper (Lenski's work in particular). We deeply appreciate the new references you provide and they have been used to better discuss our results in the Discussion section. Additionally we have included some extra citation within the context of our discussion.

We understand your view that the asymmetry in toxin susceptibility may recapitulate the results of the models exploring coexistence between toxin-susceptible and resistant types. However, since the scenario in which both species are sensible was previously unexplored, we think our results provide further insights about the range of possible phenotypes that can coexist even when competing for a common resource. This is not a minor theoretical point and we tried to provide a detailed analysis of it using mathematical, non-spatial and spatial models to test our ideas and results. The qualitative stability analysis supports the fact that considering the detoxifying species sensible to toxic changes the stability conditions with respect to the scenario considering the species as resistant, as our system's stability is conditionally stable to the magnitudes of its interactions, while a system considering the detoxifying species as resistant is not.

Regarding the metabolic tradeoff that we explore, to our knowledge, only the work from Duran-Nebreda et al. (protoorganisms (PRO) work) implemented it the way we do: the same parameter (epsilon) balances the strength of the detoxifying effect and diminishes the reproductive capacity. Despite other models diminish explicitly the fitness of the toxic degrading species, they do not link it through the detoxifying capacity as in the PRO model and ours. The importance of this parameter is shown in Fig. 3c, where the interplay between the detoxifying species and its competitor is observed. There exists optimal values of this parameter for the detoxifying species fitness, while its competitor exploits this trait also in a determinate range of epsilon values, reaching also fitness peaks.

Furthermore, the indirect positive interaction between the detoxifying species and its competitor (cooperative link) is found to be critical for cell heterogeneity. We have observed that a positive nonlinear correlation between the efficiency parameters regulating the indirect link needs to hold so that coexistence is possible. This is a further characterisation of this interaction, to our knowledge, not previously described. We have also shown (in the revised version), how the sensitivity to toxic of the detoxifying species has an influence on the possible boundaries delimiting the coexistence phase (see section XIII SM and figure below (fig. S20 SM) - the red curve is the function $\gamma_1=f(\epsilon)$ taken from equation 18 in the main text).

Finally, previous related mean field works did not support their results adding the effect of noise. Our work presents a spatial stochastic model where the main conclusions of the mean field model are maintained. Moreover, an extra domain of coexistence is found in a region not considered by the mean field model: where the competitor S_1 is less sensible to toxic than the detoxifying species S_2 . This raises an interesting result that would have not been observed if the assumption of

resistance for the detoxifying species had not been relaxed. This particular outcome requires to be further studied in future work. We have further discussed the results of our spatial stochastic model with the references provided together with additional ones.

“I think the links to evolution also need to be improved if they are going to be used to justify the significance of the work. I do not think the work has any connection to the evolution of multicellularity as there is no mechanism whereby groups of cells can give rise to new groups. I would forego the tenuous multicellularity connection in favour of the connection to microbial consortia which seems more plausible based on the model(s).”

RESPONSE: We agree with the reviewer that the connections with evolution were not clearly specified in the initial version of the manuscript. In the new version a major effort has been made to highlight the evolutionary links of the work, specially in the Discussion section.

We have better established the position of our work in terms of the social interaction between the detoxifying species and its competitor. We now show (Results new subsection in main text ‘*Nature of the social interaction between S1 and S2*’ referring to new figures and section XII in the SM) that the detoxifying species displays a cooperative trait which can either be mutually beneficial for both competitors or altruistic depending on the values of resource and toxic inputs (see section XII-A SM and Fig. S17-S18). Below we show Fig. S18 SM. The effect of the parameter epsilon has also been evaluated in these terms (see section XII-B SM and Fig. S19 SM).

In terms of its links to the origins of multicellularity, we have better put it into context. Despite no new groups of cells behaving as a new Darwinian entity are studied, an scenario predating the formation of these groups is presented. The motivation of the present work lies in a previous one where the emergence of proto-organisms with complex (multicellular) structures (Duran-Nebreda et al., PRO work) is found. The goal was to find the most fundamental causes pervading the phenotypes found in the PRO work able to form those complex structures. Moreover, let us mention that a second source of inspiration (also mentioned in the revised paper) was our experimental approach to the spatial dynamics of synthetic cooperators and their parasites, which was presented in this paper:

Amor, D.R., Montañez, R., Duran-Nebreda, S. and Solé, R., 2017. Spatial dynamics of synthetic microbial mutualists and their parasites. *PLoS computational biology*, 13(8), p.e1005689.

A clear follow-up of our current model will involve a similar mean field approximation that would allow exploring the potential phases and transitions displayed by these class of systems. Here too some surprising behavior was found in the further (not yet published) experimental results where

the “parasitic” species can turn into an additional cooperator capable of maintaining the diverse community in place against external sources of stress/toxicity.

“However, if the maintenance of diversity in a microbial community is the main focus of this work then one aspect that should be considered is what if the functional roles of types can evolve. This is a key aspect of many papers that study microbial social evolution and the maintenance of diversity/cooperation. Under the parameter regimes in the model(s) with co-existence steady states, are the steady states also evolutionary stable? In other words, if the detoxifying types could mutate to increase/decrease their detoxification rate, what happens? I think that this point is particularly relevant to the maintenance of diversity.”

RESPONSE: We sincerely appreciate the idea provided by the referee, as the evolutionary stability of any social interaction is of critical importance to be considered of interest. The references provided by the referee have been used to qualitatively discuss the stability of the proposed interaction. The results of the spatial stochastic model support the stability of the presented interaction, but no evolution of parameters was at play. We suggest that probably extra ingredients would be needed so that the interaction was evolutionarily stable, like the ones included in the PRO model. In future work we aim to fill the gap between the current work and the PRO model, in which the assessment of the evolutionary stability of the presented interaction will be of primary interest. We consider that a full, explicit evolutionary model following adaptive dynamics and changing rates would be out of the scope of this paper.